# Connective Core Structures in Cognitive Networks: The Role of Hubs

**DOI:** 10.3390/e21100961

**Published:** 2019-09-30

**Authors:** Carlos Arruda Baltazar, Maria Isabel Barros Guinle, Cora Jirschik Caron, Edson Amaro Jr., Birajara Soares Machado

**Affiliations:** 1Hospital Israelita Albert Einstein, Av. Albert Einstein 627, São Paulo 05652-900, Brazil; 2Psychology Department, Swarthmore College, 500 College Avenue, Swarthmore, PA 19081, USA; 3The Yale Stress Center, Department of Psychiatry, Yale University School of Medicine, 333 Cedar St, New Haven, CT 06510, USA; 4Inserm U894 Centre de Psychiatrie et Neurosciences, Université Paris Descartes, 1 Rue d’Alésia, 75014 Paris, France

**Keywords:** brain connectivity, complex networks, connective core

## Abstract

Complex network analysis applied to the resting brain has shown that sets of highly interconnected networks with coherent activity may support a default mode of brain function within a global workspace. Perceptual processing of environmental stimuli induces architectural changes in network topology with higher specialized modules. Evidence shows that, during cognitive tasks, network topology is reconfigured and information is broadcast from modular processors to a connective core, promoting efficient information integration. In this study, we explored how the brain adapts its effective connectivity within the connective core and across behavioral states. We used complex network metrics to identify hubs and proposed a method of classification based on the effective connectivity patterns of information flow. Finally, we interpreted the role of the connective core and each type of hub on the network effectiveness. We also calculated the complexity of electroencephalography microstate sequences across different tasks. We observed that divergent hubs contribute significantly to the network effectiveness and that part of this contribution persists across behavioral states, forming an invariant structure. Moreover, we found that a large quantity of multiple types of hubs may be associated with transitions of functional networks.

## 1. Introduction

Complex network analysis applied to neuroscience research has shed light on how the human brain efficiently transfers information using limited physical connections [1,2]. Modular processing of information occurs through segregated, specialized, and functionally modularized brain networks, and this highly efficient clustering can be observed in the visual system. For example, even though color and motion are temporally asymmetrical and autonomous modules, such that color is processed before motion [3], during the perceptual processing of a scene, information from both sensory modalities is employed simultaneously through the integration of specialized processors. Thus, cross-modal activations of specialized processors may play a role in broadcasting information to a cognitive network endorsed by hubs [4] (a set of mainly central nodes engaged in information flow and also known as the connective core).

The existence of a cognitive network that integrates information during high-level cognitive processing is predicted by the global workspace theory [5]. This theory postulates that the information processed in primary sensory areas is transmitted to a global neuronal workspace [6] that mobilizes excitatory neurons with long-range axons capable of interconnecting sensory-level and high-level processing areas [7]. The cognitive network is a locus within this global workspace and consists of interconnected brain sites [6,7]. Therefore, the global workspace establishes a framework for network cooperation, competition and state transitions for processing segregated information during effortful cognitive tasks [8]. For that reason, the cognitive network topology is posited as a dynamic state with decreased segregation of specialized sub-networks in exchange for increased brain connectivity.

The transition between segregated modular processing of specialized sub-networks to integrative processing of the cognitive networks creates a temporary metastable state of global activity [5]. Thus, determining dynamic changes in the topology of neuronal networks induced by different cognitive states enables the estimation of task effects on speed and specialization of information processing [8]. A previous study investigated the reconfiguration of network topology during the transition from resting-state to a visually-cued finger-tapping task [8]. One of the major findings of this study was that network parameters of the global topology were conserved in both states. This result suggests that, within a small-world topology, functional networks are able to support different, task-specific connectivity patterns without drastic changes in global topology. Another important finding was that motor tasks, compared to a resting-state, gave rise to higher frequency bands connecting pivotal nodes on the parietal and frontal regions. These frequencies also presented the lowest level of synchronicity of coupled oscillators in the limit between ordered and disordered behavior. The authors suggest that higher frequency networks support rapid and adaptive reconfiguration in changing environments [8].

Characterizing the effective connectivity between nodes in the cognitive network may offer insight on the gating dynamics of functional networks and workspaces. For example, nodes that are not task-specific could mark functional areas engaged in cooperative dynamics. Indeed, if information is transmitted to pivotal hubs, the networks they make up become potential components of the cognitive network. That is, the engagement of pivotal hubs could indicate that information is being integrated in a cognitive network.

## 2. Results

To study the effective connectivity of the brain using multivariate electroencephalography (EEG) data, we applied a measure called normalized transfer entropy (NTE, described in Section 4.2). In our analysis, we also proposed a betweenness-centrality-based method to classify hubs into three types (see Section 4.4 and Section 4.5): divergent hubs ($H_{D}$), which have more efferent than afferent connections; convergent hubs ($H_{C}$), which have more afferent than efferent connections; and neutral hubs ($H_{N}$), for which the amount of afferent and efferent connections are not significantly different from each other. However, for some subjects, not all hub types were present across all tasks. Thus, our analysis was conditional to the presence of the specific type of hub. Moreover, when a given type of hub was not present, we applied the Kruskal–Wallis multicomparison test, which can be used for unpaired samples, and for the other cases we used the Friedman multicomparison paired test. Finally, we proposed measures such as net entropy, homogeneity, and heterogeneity to better understand and characterize their contribution and role in network dynamics.

### 2.1. Net Entropy Individual Contribution

After measuring the effective brain connectivity matrices through the normalized transfer entropy, we applied the net entropy concept (*K*, described in Section 4.4) to evaluate the individual contribution of each type of hub during each behavioral state. In Table 1, we observe that the *K* does not differ across behavioral states for any hub type.

During cognitive tasks (ST and 2B), convergent hubs have a decreased contribution compared to their contribution during RS. Divergent hubs remain stable and maintain a large individual contribution to the effective connectivity across all behavioral states.

### 2.2. Divergent Hubs Play a Role in the Brain Network

In this study, we analyzed the impact of each hub type on the network resilience (resilience, a system’s ability to adjust its activity to retain its basic functionality when errors, failures and environmental changes occur, is a defining property of many complex systems [9,10]). As shown in Figure 1, this impact is represented by the percent variation of the characteristic path length (see Section 4.3), considering the complete network as benchmark. We observed that the withdrawal of hubs that perform a divergent role causes a greater negative impact in the characteristic path length when compared to the removal of convergent or neutral hubs. Moreover, we also noticed that these patterns are independent of behavioral state, suggesting that divergent hubs make an important contribution to the general state of connectivity.

When we compared the impact of the removal of the complete connective core or that of each hub type (see Table 2), we can see that the impact of withdrawing the connective core is approximately equal to withdrawal of all three different hub types summed together. The impact of the removal of the connective core that is associated with the removal of divergent hubs is approximately 66%.

### 2.3. The Connective Core Is Predominantly Formed by Divergent Hubs

After analyzing the contribution of each hub type, considering the characteristic path length metric, we studied the connective core composition. We also assessed the connective core’s invariance across different behavioral states. These results, displayed in Table 3, show a core structure predominantly composed of divergent hubs, regardless of the behavioral state, as shown in Figure 2.

It is important to note that the results described in Table 2 are not caused by a greater contribution of divergent hubs in connective core composition. The average percent variation in the characteristic path length when a single divergent hub was removed from the network is 34% greater than when a single neutral hub was removed in RS. For ST and 2B, these proportion were 22% and 36%, respectively. Considering the convergent hubs as benchmark, we observed the same effect for RS (38%) and 2B (14%). There was no difference between divergent and convergent average percent variation in the characteristic path length during the Stroop paradigm.

Furthermore, we used measures of homogeneity (Γ) and heterogeneity (Λ), described in Section 4.5, to quantify the sub-structure of the connective core that is preserved or not. In doing so, we observed that the connective core is partially preserved between different states, as shown in Table 4. We can observe that divergent hubs make up approximately 85% of the invariant structure between any two behavioral states. On the other hand, we also characterized hubs whose classifications change across behavioral states—highlighting a variant structure in the connective core.

### 2.4. Microstates & Hubs

We also evaluated the hubs’ dynamic behavior over time. To do so, we segmented the EEG data in 5-s moving windows with 80% overlap. Following this, we inferred a time series formed by hubs—one for each type of hub. Finally, we established a threshold parameter defined by the standard deviation of each series. Using these thresholds, we performed a microstate analysis (described in Section 4.6) triggered by the local maximum of each type of hub condition. In addition, we measured the correlation between the occurrence of these local maxima. In doing so, we observed a negative correlation between the divergent and neutral hubs across behavioral states. When we examined the correlation between $H_{D}$ and $H_{C}$, we noticed that, during the cognitive states, when compared with resting-state, the negative correlation between them increases. In Table 5, we show these results.

Paired with microstate analysis, we applied the Lempel-Ziv’s complexity measure (also described in Section 4.6) in the local time window to characterize the occurrence patterns of microstates’ sequence. We observed that, independently of the behavioral state, windows with peaks of more than one hub type presented a higher complexity than windows defined by one specific type of hub. The Friedman test was used to compare medians among the windows marked by convergent, neutral, divergent hubs and common peaks (at least two different hub types) with a significance level of 5%. For resting-state (Q=19.8; $p_{Q}=1.8674\times 10^{-4}$), the increase in complexity was approximately 17%. For Stroop (Q=20.56; $p_{Q}=1.2969\times 10^{-4}$), the increase in complexity was approximately 28%. For 2-back (Q=20.13; $p_{Q}=1.5974\times 10^{-4}$), the increase in complexity was approximately 13%.

## 3. Discussion

In the present study, we analyzed the connective core [4] of eleven subjects during effective attention [11] and working memory [12] tasks, as well as during resting-state [13]. The connective core, also called rich-club, is a substructure composed by neural hub regions that are densely interconnected and promote efficient communication and functional integration. van den Heuvel & Sporns [4] also reported a structural imbalance between incoming and outgoing projections of some brain areas, calling these regions “net receivers” and “net emitters”. This property can also affect the effective connectivity aspects, suggesting that these hubs with different behaviors can play a potential role in information flow as neural communication “sources” and “sinks”.

We proposed a similar quantified classification into three hub types: divergent hubs, convergent hubs, and neutral hubs. However, in addition to considering an imbalance to define sources (divergent hubs) and sinks (convergent hubs) nodes, we also introduced a range in which the incoming pathways are not representatively greater than the outgoing pathways or vice versa (neutral hubs). Furthermore, we evaluated the hubs’ contributions, dynamic behaviors and roles in the network. Our data reveal that divergent hubs are more effective in characterizing the network; when they are removed, there is a significant loss of effective connectivity of the whole network (see Figure 1 and Table 2). We observed that the average percent variation in the characteristic path length when a single divergent hub is removed from the network is generally greater than when any other hub type is removed, across behavioral paradigms. This effect was observed across all behavioral paradigms and supports a previous study by Kaiser et al. [14], who evaluated the robustness of brain networks when removing their nodes and edges randomly or selectively. Similar to what our data reveal, the researchers concluded that, if structures with many connections (e.g., an isolated hub) or the connective core are removed, the functional effects on the brain network should be substantial.

After understanding the contribution of the connective core and the different hubs to the network effectiveness, we analyzed the connective core’s composition. In doing so, we observed that the connective core is composed mostly of divergent hubs (see Table 3). Furthermore, we performed a tracking of the connective core across behavioral states. We observed that the connective core is composed of an invariant substructure, which is preserved between behavioral states, as well as a variant substructure that is characteristic of each paradigm (see Table 4). In addition to the divergent hubs making up most of the connective core, they also account for most of the substructure preserved between behavioral states. When looking at the variant substructure during the transition between different paradigms, certain patterns are easily noticeable: (a) half of the divergent hubs remain divergent, while the rest cease to be hubs, suggesting that divergent hubs play an integrative role; (b) neutral hubs become divergent or cease to be hubs; and (c) most convergent hubs cease to be hubs, implicating that they may be paradigm-specific hubs. These effects were described by Sporns [15], who defined central nodes as a mechanism of convergence and divergence of information flow, ensuring integrated processing. In another study, Meyer & Damasio [16] described a structural convergence zone that is responsible for the specialized process of sensory input and a structural divergence zone that integrates segregated information to build a response to sensory stimuli.

We also investigated whether the increased contribution by divergent hubs to the connective core, described above, was caused by their greater volume relative to other hubs. To accomplish that, we assessed the individual net entropies of each hub type, as described in Section 2.1. Importantly, we observed that divergent hubs contribute significantly to the effective connectivity of the global network across all behavioral states—an effect that was independent of their volume and is described in Table 1. Moreover, we observed that convergent hubs have a greater contribution during resting-state than in cognitive tasks. However, despite this larger contribution by convergent hubs, we also see that the contribution from divergent hubs is still representative during resting-state. This result can be associated to an electrophysiological signature of the default mode network (DMN); van den Heuvel & Pol [17] described the DMN as a spontaneous functional activation of anatomically distant brain regions during resting-state while the brain is waiting for a stimulus. It is well-established that the brain networks work with a balance between functional integration and segregation [18]; thus, these results suggest that the large contribution of convergent hubs would be associated with convergence zones of these anatomically distant processing modules, and that the contribution of divergent hubs characterizes this balance between functional segregation and integration.

When we investigated the dynamic behavior of different hub types, some patterns were present. We observed a “swap” relationship between neutral and divergent hubs across all paradigms, as shown in Table 5. This finding would be a consolidation of the tracking of the variant substructure over time described above, where we noted that neutral hubs have a preference to turn into divergent hubs or cease to be hubs. This preference may indicate that neutral hubs play a supporting role to divergent hubs in integrating the different sub-networks that can emerge during some behavioral states. However, we can also see that, during cognitive tasks, the “swap” between convergent and divergent hubs becomes greater than during resting-state. Analyzing the percent variation in relation to resting-state, we observed an increase of approximately 29% for ST and of more than 100% for 2B. Fair et al. [18] described that an integrative process can manage different control networks, offering a possible explanation for the increase of the observed “swap” effect during cognitive tasks.

EEG microstate topographies are thought to be electrophysiological correlates of these episodes of coherent activity proposed by the global workspace model [5,19,20,21,22]. Researchers who have tried to establish a microstate with resting-state fMRI (rsfMRI) signal have frequently observed no correlation between discrete EEG frequency bands and corresponding hemodynamic states [23,24,25]. On the other hand, microstate time courses, rather than discrete frequencies, have shown correlation with BOLD activation of distinct networks [24]. The large-scale networks described in the global workspace model have to be both stable, through the duration of a specific cognitive process, and flexible, to rapidly transition between tasks. Thus, these networks must be able to change into different patterns of connectivity at a sub-second time scale that is consistent with that of EEG microstates.

A correlation analysis of resting-state fMRI and EEG microstates to determine if a single hemodynamic network is related to one or more microstates relies on the assumption that during this “rest” period no conscious cognitive process is occurring in the brain being analyzed. However, given that participants are awake, there is no way of monitoring the amount of cognitive processes that they may engage despite the participant not being explicitly instructed to do so (i.e., to complete a specific task). When these sub-second dynamics within a period of rest are taken into account, the EEG time course correlation with distinct networks of BOLD activation makes sense [26,27,28].

Given these observed patterns, we performed an analysis of complexity of the microstates’ sequences in selected windows of EEG data, as described in Section 4.6. In doing so, we observed that in windows with common high quantities of different hub types, the microstates’ sequences complexity are higher than in windows with only one high quantity of a given hub type (see Section 2.4). Therefore, the presence of high quantities of different hub types in the same windows can be a signature of the transition between cognitive networks in a global neuronal workspace [5]; this effect was also described by Sporns [15], who asserted that the study of the connective core offers a potential substrate for understanding theories about the global neuronal workspace, high cognition, consciousness, and how disturbances on hub regions can affect integrative processes.

## 4. Materials and Methods

### 4.1. EEG Data and Tasks

For this study, we collected EEG data using a 32 channel set, with active electrodes from actiCAP and a BrainAmp amplifier (Brain Products, Gilching DE). Electrodes were placed according to the 10/20 standard system and their impedances were kept under 10 kΩ. We collected data at a sampling rate of 1 kHz. Pre-processing was done through the BrainVision analyzer 2.1 (Brain Products, Gilching DE). Data were filtered through a Butterworth bandpass (0.5–50 Hz) and a notch filter at 60 Hz. Ocular and muscular artifacts were further removed using independent component analysis and the semiautomatic correction feature of the program. Channels were referenced with respect to their average value.

We collected EEG data from eleven volunteers during a resting-state (RS) period, a “Stroop color–word test” (ST) and a “2-back working memory test” (2B). During the RS, volunteers were asked to relax and fix their gaze at a fixation cross on the screen for five minutes [13]. During the Stroop task—which has been used to assess selective attention, self-regulation, and top-down control [11,29]—participants completed three different 20-s blocks during 5 min: a congruent block, where words were printed in the color that their names represent (e.g., the word “RED” displayed in red ink); a neutral block, where words did not conflict with their color (e.g., the word “HOUSE” printed in red ink); and an incongruent block, where the colors and names of words conflicted with each other (e.g., the word “RED” printed in blue ink). Finally, participants also completed a 2B task—a well-established cognitive test that evaluates working memory. In this task, during a 7-min time span, subjects were exposed to a sequence of visual stimuli and had to indicate whether the stimulus they were currently looking at matched the one shown two trials prior [12].

### 4.2. Normalized Transfer Entropy

To model the effective connectivity, we used a directional measure called normalized transfer entropy (NTE), proposed by Shovon et al. [30]. This measure builds upon the transfer entropy concept created by Schreiber [31]. NTE incorporates nonlinear, dynamic connections, and directional properties to determine if a pattern of brain activation is dependent on another pattern of activity and not on its own past activity. Thus, transfer entropy characterizes the information flow between two signals, and is defined by
(1)$$TE_{Y\to X}=\sum_{t}p\left(x_{t+1},x_{t},y_{t}\right)\log_{2}\frac{p(x_{t+1}|x_{t},y_{t})}{p(x_{t+1}|x_{t})}\;,$$
where $p(x_{t+1},x_{t},y_{t})$ is the joint probability among $x_{t+1}$, $x_{t}$ and $y_{t}$. Moreover, we define the deviation from causal independence considering the generalized Markov property $p\left(x_{t+1}|x_{t},y_{t}\right)=p\left(x_{t+1}|x_{t}\right)$. When there is no causal relationship between the signals, TE goes to zero. TE is an asymmetric measure, thus $TE_{Y\to X}\neq TE_{X\to Y}$, and characterizes information about $x_{t+1}$ from the observations $x_{t}$ and $y_{t}$.

However, the finite size and nonstationarity of EEG data introduces uncertainty on the TE measurement. To obtain a suitable estimate, Shovon et al. [30] proposed two additional steps that increase accuracy. These two steps consist of subtracting the mean value of $TE_{\tilde{Y}\to X}$ from $TE_{Y\to X}$ (where $\tilde{Y}$ is a surrogate randomization of the *Y* signal) and normalizing the measure by the conditional entropy of $x_{t+1}$ and $x_{t}$, $H\left(x_{t+1}|x_{t}\right)=-\sum_{x_{t+1},x_{t}}p\left(x_{t+1},x_{t}\right)\log_{2}\left(p\left(x_{t+1},x_{t}\right)/p\left(x_{t}\right)\right)$, where $p(x_{t+1},x_{t})$ is the joint probability of $x_{t+1}$ and $x_{t}$. Therefore, the NTE is defined as
(2)$$NTE_{Y\to X}=\frac{TE_{Y\to X}-\langle TE_{\tilde{Y}\to X}\rangle }{H(x_{t+1}|x_{t})}\;,$$
where 〈·〉 indicates the average over *n* random realizations. Here, we adopted n=30. In Figure 3, we show the connectivity matrices inferred by NTE for all behavioral states.

### 4.3. Order Parameters

To study the centrality properties of networks and define their hubs, we used a measure based on the shortest path length concept, called betweenness centrality (BC). This measure was used alongside the method for hub definition proposed by da Silva et al. [32]. For two given nodes *k* and *j* from the set of nodes *N*, the shortest path length is defined by $d_{kj}=\sum_{a_{uv}\in g_{k↔j}}a_{uv}$, where $a_{uv}$ belongs to a connectivity matrix and $g_{k↔j}$ is the shortest path length between the nodes *i* and *j* [33]. Thus, BC is the fraction of all shortest path lengths that a given node participates in, expressed by
(3)$$BC_{i}=\frac{1}{\left(n-1)(n-2\right)}\sum_{k\in N}\sum_{j\in N}\frac{\#d_{kij}}{\#d_{kj}},\mathrm{for}\;k\neq j,\;k\neq i,\;j\neq i\;,$$
where $\#d_{kj}$ is the number of shortest paths between *k* and *j*, $\#d_{kij}$ is the number of shortest paths between *k* and *j* that pass through *i*, and *n* is the number of nodes.

Following this measure, we applied the method of da Silva et al. [32] to identify hubs from nodes. We used a left-sided Mann–Whitney test with significance level of 5% to compare the BC value of each node with all remaining nodes. The node that had a statistically higher BC value than the others was classified as a hub.

Moreover, we applied another order parameter called characteristic path length, denoted by *ℓ*, and defined by the average shortest path length between of all pairs of nodes:(4)$$\ell =\frac{1}{n(n-1)}\sum_{k\in N}\sum_{j\in N,k\neq j}d_{kj}\;.$$

### 4.4. Divergent, Convergent and Neutral Hubs

In addition to identifying the hubs in the network, we also proposed a classification method to study in detail the role of these nodes in the effective connectivity of the brain. This method uses the net normalized transfer entropy, denoted by
(5)$$K_{i}=\sum_{j=1}^{N}NTE_{i\to j}-\sum_{m=1}^{N}NTE_{m\to i}\;,$$
where $\sum_{j=1}^{N}NTE_{i\to j}$ is the output degree (OD) and $\sum_{m=1}^{N}NTE_{m\to i}$ is the input degree (ID). If a given hub has an OD 10% higher than its ID, this node is classified as a divergent hub ($H_{D}$). If a given hub’s ID is 10% higher than its OD, this node is classified as a convergent hub ($H_{C}$). When these two conditions are not satisfied, the hub is classified as a neutral hub ($H_{N}$), as illustrated in Figure 4.

### 4.5. Homogeneity and Heterogeneity Measures

To work with the global workspace concept proposed by Dehaene et al. [5] on complex networks metrics, it was necessary to track back the hubs’ behaviors across tasks. To track a hub’s role in the connective core during each behavioral state, we developed a measure, which we call the homogeneity measure, that quantifies the stable percent hub composition of the connective core during tasks. Our homogeneity measure is defined by:(6)$$\Gamma =\frac{2n(A\cap B)}{n(A)+n(B)}\;,$$
where *A* is a given set of different hub types of a specific task, *B* is the hub type set of a second task and *n* is the cardinality of each hub set. In addition, we also developed a measure to calculate the variable percent hub composition of the connective core during tasks, called the heterogeneity measure and defined by
(7)$$\Lambda =\frac{n\left(A^{\prime }\right)+n\left(B^{\prime }\right)}{n(A)+n(B)}\;,$$
where $A^{\prime }$ is the complementary set of *A* and $B^{\prime }$ is the complementary set of *B*.

### 4.6. Brain Microstates and Hubs

Microstate analysis is one of the several methods that make use of EEG recordings. Microstates consist of a series of predictable, quasi-stable topographical states of electrical potentials lasting on average 100 ms (see Figure 5). This technique is a useful tool to analyze the function of large-scale brain networks because the simultaneous activity of cortical regions generates the topographical potential maps/microstates. Hence, a change in topography—a transition from one microstate to another—corresponds to an electrical change in potential, which in turn is interpreted as a change in the activation of functional networks [34,35].

Attempts have been made to determine if EEG microstates are electrophysiological signatures of hemodynamic signals in rsfMRI [24,25]. The global workspace theory, postulated by Baars [6], and later elaborated on by Dehaene & Naccache [7], highlights that top-down attentional mechanisms have an influence on the global neural representation of information. In essence, this means that the synchronicity of neural activity is a requirement for global workspace activation, such that synchrony and connectivity become essential features of global workspace engagement associated with intentional actions. That is, while several networks may be active during cognitive tasks, different networks will be preferentially activated through the duration of that task.

Thus, to establish a parallel between the global workspace theory and the invariant structures formed by hubs exposed by the present work, we segmented the EEG data into mobile windows of five seconds with an 80% overlap. Microstate analysis was triggered during windows that presented peaks of quantity one standard deviation above the mean for a given hub type. Moreover, in these windows, we applied an adaptation of complexity measure proposed by Lempel & Ziv [36] for quaternary data to assess microstates time course patterns. The complexity measure allowed us to quantify and study microstates time course patterns.

## 5. Conclusions

In this study, we analyzed the connective core network across three behavioral conditions: resting-state, Stroop and 2-back paradigms. We observed that divergent hubs are more effective in characterizing the network, independent of the fact that these hubs are the most abundant type in the the connective core. In addition, we noticed that the connective core is composed of both an invariant substructure that is preserved between behavioral states, and a variant substructure that is characteristic of each cognitive task and the resting-state. The constructs proposed here can be used to analyze transitions in brain states associated with task components and/or resting-states, playing a relevant role in the balance between functional segregation and integration. Moreover, we observed that the effectiveness of convergent hubs during resting-state may be associated with anatomically distant modules described in the default mode network. Furthermore, the prevalence of a specific hub type supports the existence of well-established functional networks, while the transitions observed characterize an increased expression of different hub types.

This description may bring light to the role of functional brain networks across different contexts, from cognitive processes in healthy subjects to disordered neural events. Thus, studying the connective core offers a potential substrate for understanding theories about the global neuronal workspace, high cognition, consciousness, and how disturbances on hub regions can affect integrative processes.

## Figures and Tables

**Figure 1 entropy-21-00961-f001:**
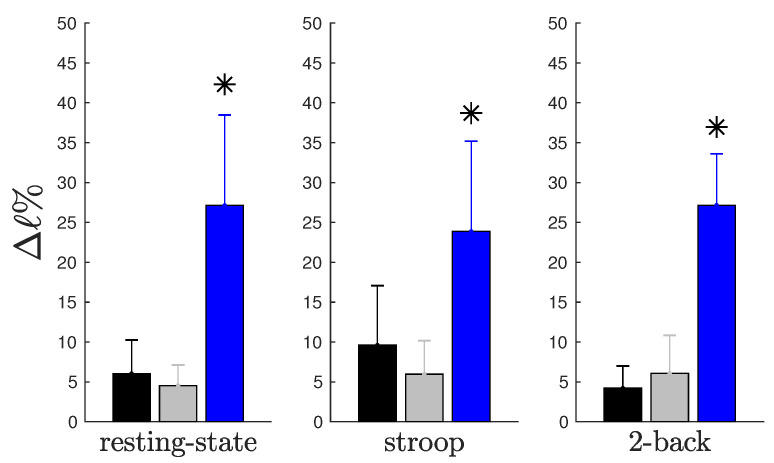
Graphical representation of Table 2. Average percent impact on characteristic path length (ℓ) when each type of hub was removed. The blue bars represent the divergent hubs’ impact, the gray bars represent the convergent hubs’ impact and the black bars represent the neutral hubs’ impact. Each graph shows the impact of three different hub types during resting-state (RS), Stroop (ST) and 2-back (2B), respectively. We found differences in impact between the removal of divergent hubs compared to that of the two other types in the RS (K=18.13; $p_{K}=1.1586\times 10^{-4}$), ST (K=12.16; $p_{K}=0.0023$) and 2B (K=17.88; $p_{K}=1.3130\times 10^{-4}$) paradigms. We applied the Kruskal–Wallis test with a significance level of 5%.

**Figure 2 entropy-21-00961-f002:**
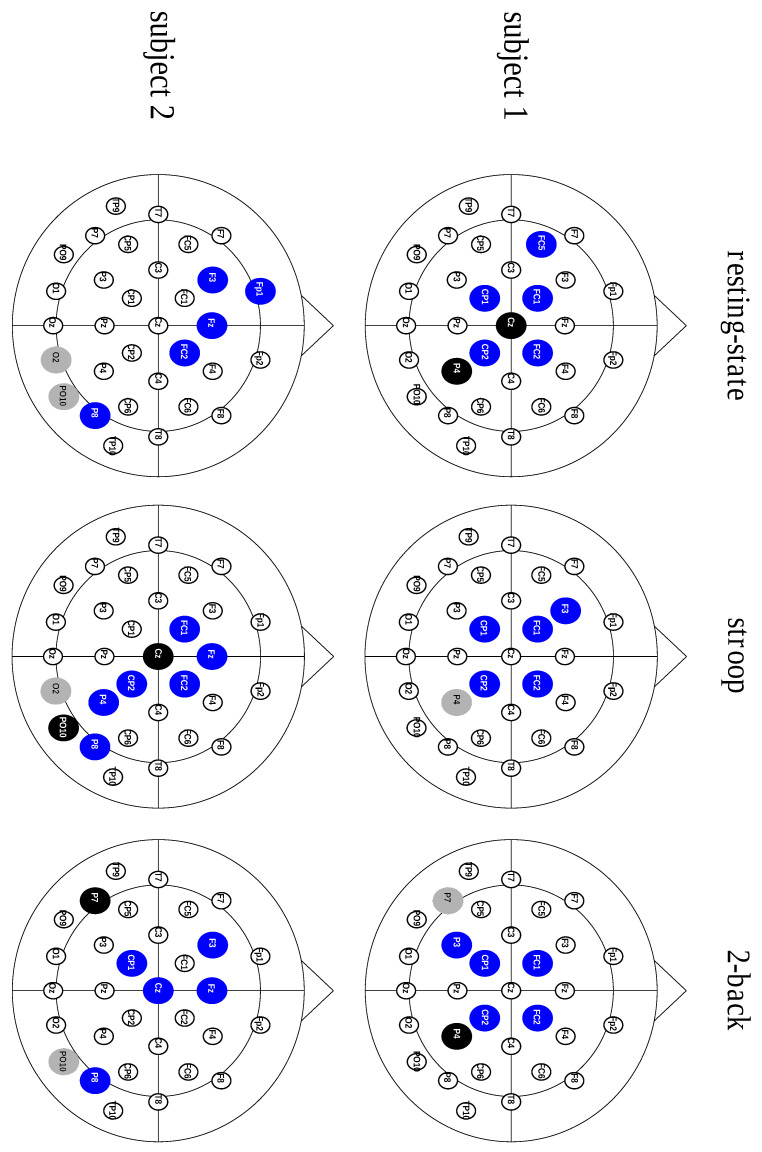
Scalp representation of three hub types across the behavioral states from two different subjects. The blue nodes illustrate the divergent hubs, the black nodes represent the neutral hubs and the gray nodes show the convergent hubs.

**Figure 3 entropy-21-00961-f003:**

Connectivity matrices inferred by the NTE (see Section 4.2) from EEG data of one subject. For each connectivity matrix, the *x*-axis represents the information generator channels and the *y*-axis represents the information receiver channels.

**Figure 4 entropy-21-00961-f004:**
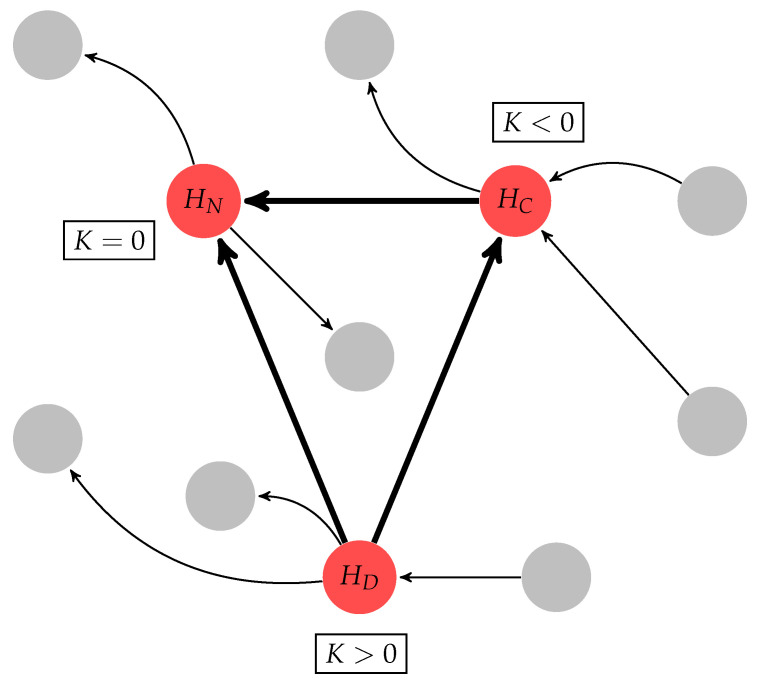
Graphical representation of the hub classification, where the arrows in bold illustrate the connective core, gray nodes represent non-hub nodes, and red nodes represent hubs. The connective core is classified into three types of hubs using the *K* measure (see Section 4.4): divergent hub ($H_{D}$); neutral hub ($H_{N}$); and convergent hub ($H_{C}$).

**Figure 5 entropy-21-00961-f005:**
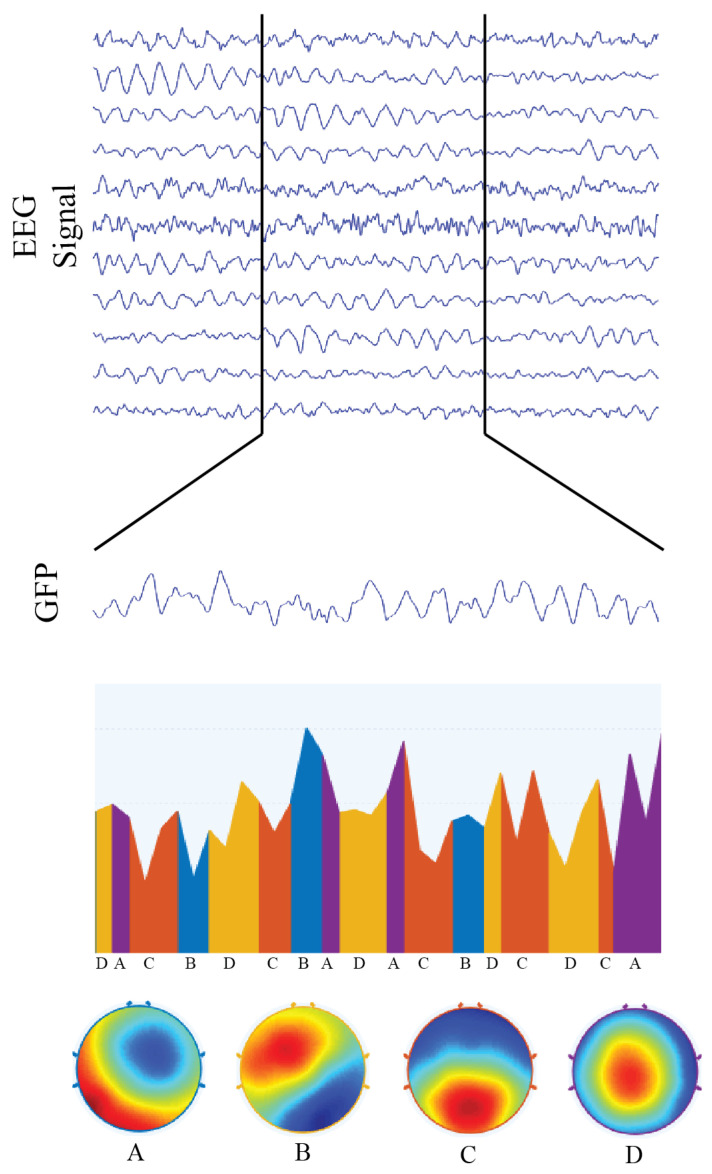
Representation of microstate analysis for 2500 ms of multivariate EEG signal condensed into a single “global field signal” $GFP_{t}=\sqrt{\sum_{i=1}^{N_{\mathrm{ch}}}{\left({\mathrm{EEG}}_{i,t}-\langle {\mathrm{EEG}}_{t}\rangle \right)}^{2}/N_{\mathrm{ch}}}$, from which electrical potential topographies were calculated and then classified into four functional networks, called microstates. We also illustrate the transitions of microstates over time, as well as each microstate topography. Microstate “A” corresponds to the topographic representation of the auditory functional network; microstate “B” is associated with the visual functional network; microstate “C” is associated with the default mode network; and microstate “D” corresponds to the topographic representation of the attentional functional network.

**Table 1 entropy-21-00961-t001:** Mean net entropy (*K*) of each behavioral state. Mean entropies represent the average contribution from individual hubs. Divergent hubs are more stable across the behavioral states. In addition, in Stroop and 2-back tasks, divergent hubs also have a more representative contribution when compared to that of convergent hubs. During resting-state, however, the contribution of convergent hubs is more representative. $\%\Delta K_{ST}/K_{RS}$ and $\%\Delta K_{2B}/K_{RS}$ indicate the percent variation of *K* (with resting-state as reference) in Stroop and 2-back tasks, respectively.

Hub Classification	Resting-State	Stroop	2-Back	$\%\Delta K_{\mathrm{ST}}/K_{\mathrm{RS}}$	$\%\Delta K_{2B}/K_{\mathrm{RS}}$
$H_{D}$	0.2986	0.3490	0.3160	16.8788%	5.8272%
$H_{C}$	−0.7293	−0.2440	−0.2390	66.5433%	67.2288%
$H_{N}$	0.0171	0.0210	0.0170	22.8070%	−0.5848%

**Table 2 entropy-21-00961-t002:** Average percent impact on characteristic path length when the connective core or each hub type was removed.

Hub Classification	Resting-State	Stroop	2-Back
Network minus core	39.65%	40.18%	39.36%
Network minus $H_{D}$	27.13%	23.86%	27.15%
Network minus $H_{C}$	4.53%	5.97%	6.07%
Network minus $H_{N}$	6.04%	9.60%	4.21%

**Table 3 entropy-21-00961-t003:** Percent average in the connective core composition. Divergent hubs represent the greater part of central nodes in RS (Q=14.37; $p_{Q}=7.5708\times 10^{-4}$), ST (Q=12.7; $p_{Q}=0.0017$) and 2B (Q=17.71; $p_{Q}=1.4286\times 10^{-4}$) paradigms. We applied the paired Friedman test considering a significance level of 5%. Multiple comparison tests indicated that the divergent hubs make up the greatest portion in the connective core composition.

Hub Classification	Resting-State	Stroop	2-Back
$H_{D}$	64.61%	65.82%	72.91%
$H_{C}$	13.47%	14.68%	14.40%
$H_{N}$	21.92%	19.51%	12.69%

**Table 4 entropy-21-00961-t004:** Percent average variation between hub types in two behavioral states. Divergent hubs are more preserved and point out to a shared network that remains activated.

Connective Core	Hub Change	RS∩ST	RS∩2B	ST∩2B
Invariant structure	$H_{D}\to H_{D}$	32.94%	34.03%	35.64%
	$H_{C}\to H_{C}$	01.14%	02.31%	02.27%
	$H_{N}\to H_{N}$	05.84%	04.16%	00.96%
*Total preserved*		39.92%	42.18%	38.87%
Variant structure	$H_{D}\to H_{N}$	02.61%	01.21%	02.81%
	$H_{D}\to H_{C}$	00.00%	01.30%	00.00%
	$H_{D}\to N_{H}$	26.01%	25.92%	28.39%
	$H_{C}\to H_{D}$	01.30%	03.93%	01.65%
	$H_{C}\to H_{N}$	01.14%	00.00%	01.40%
	$H_{C}\to N_{H}$	10.57%	06.99%	09.66%
	$H_{N}\to H_{D}$	08.82%	10.52%	05.86%
	$H_{N}\to H_{C}$	01.40%	03.32%	03.34%
	$H_{N}\to N_{H}$	08.24%	06.31%	08.02%
*Total*		100.00%	100.00%	100.00%

**Table 5 entropy-21-00961-t005:** Averaged values (considering all subjects) of the correlation between the presence of two given hub types over the time. %ΔST/RS and %Δ2B/RS indicate the percent variation of the correlation (considering the resting-state as reference) in Stroop and 2-back, respectively.

Correlation	RS	ST	2B	%ΔST/R *S*	%Δ2B/RS
$H_{N}\times H_{D}$	−0.3323	−0.3576	−0.3022	−7.6113%	9.0586%
$H_{N}\times H_{C}$	−0.2135	−0.1747	−0.2026	18.1746%	5.1016%
$H_{C}\times H_{D}$	−0.0987	−0.1281	−0.2013	−29.7501%	−103.8972%

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
