# Peer review of "Connective Core Structures in Cognitive Networks: The Role of Hubs"

_entropy, 2019, doi:10.3390/e21100961_

Round 1

Reviewer 1 Report

The paper proposed a novel analysis of a brain functional network derived from transfer entropy measures of EEG data.  In particular, the authors propose to use the functional network to define "divergent", "convergent" and "neutral" hubs. The primary findings, as I understand them, is that 1) most hubs are divergent, and 2) removing divergent hubs impacts the characteristic path length of the functional network much more than removing other kinds of hubs.

I found the paper relatively well written, except for the Methods section which needs some polishing of the English.

I have two major criticisms, which can be addressed by additional analyses:

1) The authors show that removing the divergent hubs impacts the characteristic path length much more than the other hubs. In doing this analysis, I believe that all divergent hubs were removed at once (it should be more clearly stated if this is the case or not).  However, it makes sense that removing more nodes would have a greater impact on the characteristic path length (possibly in a very nonlinear way) -- and if I understand correctly, there are many more divergent nodes than other kinds.  I believe the authors should include a table or figure showing how the average path length changes when a single divergent hub vs a single convergent hub vs a single neutral hub is removed (averaged across hubs of each type). This would allow the reader to compare the relative impact of the hubs in a more fair manner.

2) I am not convinced that the predominance of divergent hubs is not an artifact of the complicated measures and analysis steps used by the authors (e.g., not an artifact of the transfer entropy measure used). I believe some kind of simple null model or control is needed to support this finding. For instance, the authors could carry out their analysis on a established neural mass model (David and Friston Neuroimage 2003) or perhaps a simple autoregressive process, so as to show that divergent hubs are not uncovered when the underlying causal model does not have any underlying "divergent hub" structure.

Other comments:

* Transfer entropy for EEG data is usually performed in the frequency domain. Moreover, it has been shown that the pattern of information flow in the brain differs strongly among different brain frequencies (Hillebrand  et al PNAS 2016). I believe the manuscript would be much stronger if the findings could be replicated for phase-based TE performed on band-passed data.

* The definition of "connective core" is actually not stated before it is used. I believe it refers to the set of all hubs. If so, this should be emphasized clearly. 

* I believe it would be very insightful if the authors could present a scalp visualization, showing the location of the divergent, convergent, and neutral hubs.

* I believe Figure 1 and Table 1 show the same data. If this is so, this should be more clearly stated, and they should be placed next to each other.

* Typo in Line 89 and Line 223 ("their" -> "they") 

Author Response

August 12, 2019

Response to reviewers for the manuscript titled: Connective Core Structures in Cognitive Networks: the Role of Hubs.

Dear editor,

We are pleased to resubmit a new version of our manuscript to Entropy (Information Theory and Complexity Science Approaches to Health Conditions and Cognitive Decline - Special Issue). The manuscript was restructured following the reviewers’ comments (changes are highlighted in the document).

We thank them for their suggestions to improve our manuscript structure and to clarify several definitions and results.

Sincerely,

Carlos Arruda Baltazar

*****************************************************************************

Reviewer 1: 

1) The authors show that removing the divergent hubs impacts the characteristic path length much more than the other hubs. In doing this analysis, I believe that all divergent hubs were removed at once (it should be more clearly stated if this is the case or not).  However, it makes sense that removing more nodes would have a greater impact on the characteristic path length (possibly in a very nonlinear way) -- and if I understand correctly, there are many more divergent nodes than other kinds. I believe the authors should include a table or figure showing how the average path length changes when a single divergent hub vs a single convergent hub vs a single neutral hub is removed (averaged across hubs of each type). This would allow the reader to compare the relative impact of the hubs in a more fair manner.

Response 1: To address the suggestion made by reviewer 1, we added a table (Tab. 3) in section 2.3 (Divergent hubs play a role in the brain network) where we removed the hubs individually and measured their impact on the characteristic path length. As recommended, this highlights the importance of the divergent hubs in the network.

2) I am not convinced that the predominance of divergent hubs is not an artifact of the complicated measures and analysis steps used by the authors (e.g., not an artifact of the transfer entropy measure used). I believe some kind of simple null model or control is needed to support this finding. For instance, the authors could carry out their analysis on an established neural mass model (David and Friston Neuroimage 2003) or perhaps a simple autoregressive process, so as to show that divergent hubs are not uncovered when the underlying causal model does not have any underlying "divergent hub" structure.

Response 2: Thank you for making this suggestion. The methodology we adopted to classify hubs was the same for all hub types. Even though we see the benefits of using an autoregressive process, the suggested process measures linear properties – and the properties we were assessing are nonlinear. Since the properties analyzed were nonlinear, we kept our original measures. As we presented in an earlier work (Measures for brain connectivity analysis: nodes centrality and their invariant patterns. EPJ ST, v. 226, n. 10, p. 2235-2245, 2017), different measures do not necessarily characterize the same properties, therefore, we opted for the transfer entropy measure because it is a directional measure based on probabilistic properties. Furthermore, the goal of this paper was to observe dynamic phenomena of neural connectivity in different behavioral states rather than to compare the different measures of inferring connectivity, which has been done in the paper previously mentioned, we thought that including an additional measure may divert from our original goal.

3) Transfer entropy for EEG data is usually performed in the frequency domain. Moreover, it has been shown that the pattern of information flow in the brain differs strongly among different brain frequencies (Hillebrand et al. PNAS 2016). I believe the manuscript would be much stronger if the findings could be replicated for phase-based TE performed on band-passed data.

Response: For this paper, we opted to keep the original measure of transfer entropy proposed by Schreiber (Measuring information transfer. Physical review letters, v. 85, n. 2, p. 461, 2000.). However, we found the reviewers’ suggestion extremely interesting and plan on extending our research to the frequency domain in our future papers, incorporating the analyses described in Hillebrand et al. PNAS 2016. 

4) Other comments:

* The definition of "connective core" is actually not stated before it is used. I believe it refers to the set of all hubs. If so, this should be emphasized clearly. 

Response: Addressing this suggestion, we added a footnote with the connective core’s definition when it is first mentioned (page 1) in the paper.

* I believe it would be very insightful if the authors could present a scalp visualization, showing the location of divergent, convergent, and neutral hubs.

Response: We added a figure (Fig.2), in section 2.3 (The connective core is predominantly formed by divergent hubs) as a graphical representation of the hubs on the scalp of two subjects.

* I believe Figure 1 and Table 1 show the same data. If this is so, this should be more clearly stated, and they should be placed next to each other.

Response: We added a transition phrase that now clearly points out that Figure 1 and Table 1 show the same data, and we also moved them next to each other.

* Typo in Line 89 and Line 223 ("their" -> "they") 

Response: The typos should no longer be present.

Reviewer 2 Report

This is an interesting paper, where the network of transfer entropy is exploited to mine hubs in cognitive networks of the brain. The paper is well written and contains material which deserves publication in Entropy.

As a minor recommendation, I would suggest the authors to show also results in terms of hubs of the functional connectivity (Pearson correlations), are the hubs of transfer entropy networks the same as those from functional connectivity? Do they provide complementary information about the neural dynamics?

Author Response

August 12, 2019

Response to reviewers for the manuscript titled: Connective Core Structures in Cognitive Networks: the Role of Hubs.

Dear editor,

We are pleased to resubmit a new version of our manuscript to Entropy (Information Theory and Complexity Science Approaches to Health Conditions and Cognitive Decline - Special Issue). The manuscript was restructured following the reviewers’ comments (changes are highlighted in the document).

We thank them for their suggestions to improve our manuscript structure and to clarify several definitions and results.

Sincerely,

Carlos Arruda Baltazar

************************************************************************************************

Reviewer 2: 

1) As a minor recommendation, I would suggest the authors to show also results in terms of hubs of the functional connectivity (Pearson correlations), are the hubs of transfer entropy networks the same as those from functional connectivity? Do they provide complementary information about the neural dynamics?

Response: Thank you for pointing this out, we corrected/clarified this concept in the text (see line 122).

Round 2

Reviewer 1 Report

I appreciate the changes the authors made to the manuscript, which I think in general greatly clarifies things.

However, now that I understand the results better, I am quite confused.  Figure 2 shows that Subject 1 had 0 (!) convergent hubs on one condition and 0 neutral hubs on another condition, while Subject 2 had 0 neutral hubs on one condition. I am now unsure how to understand many of the results presented in the paper. For example, what do the entries in Table 4 ("Average percent impact in characteristic path length when a single hub of a given type was removed from network.") mean?  Isn't it actually impossible to remove a single hub from several of the conditions / subjects, because they do not appear for some conditions / subjects? I have similar worries about Table 2, how can one measure the effect of removing a set of nodes which do not exist? Etc.

I apologize if I am misunderstanding something fundamental.

Author Response

August 20, 2019

Response to reviewers for the manuscript titled: Connective Core Structures in Cognitive Networks: the Role of Hubs.

Dear editor,

We are pleased to resubmit a new version of our manuscript to Entropy (Information Theory and Complexity Science Approaches to Health Conditions and Cognitive Decline - Special Issue). The manuscript was restructured following the reviewers’ comments (changes are highlighted in the document).

We thank them for their suggestions to improve our manuscript structure and to clarify several definitions and results.

Sincerely,

Carlos Arruda Baltazar

*********************************************************************************************************

Reviewer 1: 

1) I appreciate the changes the authors made to the manuscript, which I think in general greatly clarifies things.

However, now that I understand the results better, I am quite confused.  Figure 2 shows that Subject 1 had 0 (!) convergent hubs on one condition and 0 neutral hubs on another condition, while Subject 2 had 0 neutral hubs on one condition. I am now unsure how to understand many of the results presented in the paper. For example, what do the entries in Table 4 ("Average percent impact in characteristic path length when a single hub of a given type was removed from network.") mean?  Isn't it actually impossible to remove a single hub from several of the conditions / subjects, because they do not appear for some conditions / subjects? I have similar worries about Table 2, how can one measure the effect of removing a set of nodes which do not exist? Etc.

I apologize if I am misunderstanding something fundamental.

Response 1: First, we would like to thank the reviewer for the observations made. We reviewed our methodology and re-ran some of our analyses taking the comments into account. We also added an introductory paragraph in the results section to clarify our methodology (lines 55 to 66). After re-running the analyses, our patterns still remain; the only change is addressed in lines 92 to 98 of section 2.3 - reinforcing the individual contributions of divergent hubs. Consequently, we also made changes to the discussion (lines 140 to 143).

Round 3

Reviewer 1 Report

The authors have responded to my concerns.